# Discovery of Novel Diarylamide *N*-Containing Heterocyclic Derivatives as New Tubulin Polymerization Inhibitors with Anti-Cancer Activity

**DOI:** 10.3390/molecules26134047

**Published:** 2021-07-02

**Authors:** Xu Liu, Xiao-Jing Pang, Yuan Liu, Wen-Bo Liu, Yin-Ru Li, Guang-Xi Yu, Yan-Bing Zhang, Jian Song, Sai-Yang Zhang

**Affiliations:** 1Key Laboratory of Advanced Drug Preparation Technologies (Ministry of Education), Institute of Drug Discovery & Development, School of Pharmaceutical Sciences, Zhengzhou University, Zhengzhou 450001, China; lx012496@163.com (X.L.); summer_pxj@163.com (X.-J.P.); liuyuan9720@163.com (Y.L.); 17698567571@163.com (W.-B.L.); ygx19990110@126.com (G.-X.Y.); zhangyb@zzu.edu.cn (Y.-B.Z.); 2School of Basic Medical Sciences, Zhengzhou University, Zhengzhou, 450001, China; 17603868158@163.com

**Keywords:** tubulin, CA-4, vicinal diaryl, *N*-containing heterocyclics, anti-proliferative activity, colchicine binding site

## Abstract

Tubulin has been regarded as an attractive and successful molecular target in cancer therapy and drug discovery. Vicinal diaryl is a simple scaffold found in many colchicine site tubulin inhibitors, which is also an important pharmacophoric point of tubulin binding and anti-cancer activity. As the continuation of our research work on colchicine binding site tubulin inhibitors, we designed and synthesized a series of diarylamide *N*-containing heterocyclic derivatives by the combination of vicinal diaryl core and *N*-containing heterocyclic skeletons into one hybrid though proper linkers. Among of these compounds, compound **15b** containing a 5-methoxyindole group exhibited the most potent inhibitory activity against the tested three human cancer cell lines (MGC-803, PC-3 and EC-109) with IC_50_ values of 1.56 μM, 3.56 μM and 14.5 μM, respectively. Besides, the SARs of these compounds were preliminarily studied and summarized. The most active compound **15b** produced the inhibition of tubulin polymerization in a dose-dependent manner and caused microtubule network disruption in MGC-803 cells. Therefore, compound **15b** was identified as a novel tubulin polymerization inhibitor targeting the colchicine binding site. In addition, the results of molecular docking also suggested compound **15b** could tightly bind into the colchicine binding site of β-tubulin.

## 1. Introduction

Tubulin is an important component of the cytoskeleton and plays a vital role in the process of maintaining normal cell formation, cell mitosis, signal transduction and material transportation [1,2]. Tubulin has been regarded as an attractive and successful molecular target in cancer therapy and drug discovery [3]. Tubulin-targeting agents with excellent anti-cancer activity such as paclitaxel/Taxol, vincristine and vinblastine have been successfully used in clinical treatment [4]. However, owing to the poor water solubility, drug resistance or side effects of clinical used tubulin inhibitors [5,6,7], it is necessary to develop novel tubulin inhibitors [6]. However, colchicine site tubulin inhibitors have received extraordinary attention in the recent years, which could largely overcome the above drawbacks and have more therapeutic advantages over other binding sites tubulin inhibitors [8,9]. In addition, no colchicine site tubulin inhibitors have not been approved for clinical use; therefore, it is necessary to develop novel colchicine site tubulin inhibitors.

Vicinal diaryl is a simple scaffold found in many colchicine site tubulin inhibitors, which is also an important pharmacophoric point of tubulin binding and anti-cancer activity [10,11,12,13]. Natural colchicine site tubulin inhibitor combretastatin A-4 (**CA-4**) is a 1,2-diarylethylene analog, which exhibits potent inhibitory potency against many cancer cells including multidrug resistant cancer cells and could effectively inhibit the polymerization of tubulin [14]. However, due to the poor water solubility, low bioavailability and unstable cis double bond of **CA-4 [12,15]**, its clinical application is limited; therefore, clinical work with **CA-4** was carried out with the corresponding water-soluble phosphate prodrug salt (**CA-4P**) [16]. Therefore, many groups have reported novel colchicine site tubulin inhibitors with vicinal diaryl cores base on the structural optimizations of **CA-4 [17,18,19,20,21,22,23,24,25]**. Cushman et al. reported a novel tubulin inhibitor benzyl aniline **1** in which the olefinic bridge of the **CA-4** was replaced by an aminomethylene hydrochloride moiety. Compound **1** retained significant anti-proliferative activity and inhibitory potency of tubulin polymerization [23]. Romagnoli et al. replaced the usual ethylene bridge of the led **CA-4** with a five membered heterocyclic ring to obtain 1,5-disubstituted 1,2,4-triazole **2**. Compound **2** exhibited potent inhibitory effect on tubulin polymerization (IC_50_ = 2.3 μM), and significantly suppressed the growth of human cancer cells at nano-molar levels [24]. Meegan′s group reported a class of diaryl-β-lactam derivatives which contained the β-lactam ring system in place of the ethylene bridge of the led **CA-4 [25]**. Compound **3** displayed sub nano-molar inhibitory activity against breast cancer cells MCF-7 and MDA-MB-231 (IC_50_ = 34 and 78 nM, respectively) together with significant inhibition of tubulin polymerization (Figure 1). Therefore, tubulin inhibitors bearing the vicinal diaryl moiety may be favorable for the interactions with tubulin.

*N*-containing heterocyclic skeleton is one of most attractive frameworks in bioactive compounds with strong pharmacological significance [26], especially the anti-cancer ability. For example, indole [27,28,29,30,31,32], indoline [33,34,35] and tetrahydroquinoline derivatives [36,37,38] have also been used to design novel colchicine binding site tubulin inhibitors with potent anti-cancer activity. Chalcone indole derivative **4** potently inhibited cancer cell growth with IC_50_ values ranging from 0.22 to 1.80 μM [30]. Compound **4** induced cell cycle arrest in G2/M phase and effectively inhibited the polymerization of tubulin. Quinoline-indole derivative **5** as an anti-tubulin agent targeting the colchicine binding site showed effective inhibition effect against tested cancer cell lines with IC_50_ values ranging from 2 to 11 nM together with significant in vitro inhibition of tubulin polymerization (IC_50_ = 2.09 μM) [31]. Benzimidazole-indole derivative **6** exhibited potent inhibitory effects on the growth of cancer cells with an average IC_50_ value of 50 nM together with significant in vitro inhibition of tubulin polymerization (IC_50_ = 2.52 μM) [32]. The 7-Aroyl-aminoindoline-1-sulfonamide **7** potently inhibited the tubulin polymerization (IC_50_ = 1.1 μM) by binding to colchicine binding site. Compound **7** effectively suppressed the growth of KB, MKN45, H460, HT29 and TSGH cells with IC_50_ values ranging from 8.6 nM to 10.8 nM [35]. Tetrahydroquinoline-pyrimidine [38] showed strong anti-proliferative activity (IC_50_ values ranging from 5.6 to 18.3 nM) via tubulin polymerization inhibition and potently inhibited tumor growth in an A375 melanoma xenograft model (Figure 2).

Based on the above findings and as the continuation of our group work on colchicine binding site tubulin inhibitors, we designed and synthesized a series of diarylamide *N*-containing heterocyclic derivatives by the combination of vicinal diaryl core and *N*-containing heterocyclic skeleton into one hybrid though proper linkers (Figure 3). Among of these compounds, compound **15b** was identified as a tubulin inhibitor with potent anti-cancer activity.

## 2. Results and Discussion

### 2.1. Chemistry

As shown in Scheme 1, all the target compounds **15a**–**15h** and **19a**–**19g** were synthesized by starting from commercially available 4-methoxybenzylchloride **9** and 3,4,5-trimethoxyaniline **10**. The 4-Methoxybenzylchloride **9** reacted with the 3,4,5-trimethoxyaniline **10** in the presence of K_2_CO_3_ in DMF to give compound **11**. Then compound **11** reacted with chloroacetyl chloride to afford compound **13** in DMF. Substitution reactions between compound **13** with indoles, indolines or tetrahydroquinolines **14** in the presence of K_2_CO_3_ in acetonitrile gave compounds **15a**–**h**. In the synthesis of another series of compounds, compounds **14** reacted with 3-bromopropyne **16** in the presence of K_2_CO_3_ in acetonitrile to obtain compound **17**. Substitution reaction between compound **15** with sodium azide in the presence of K_2_CO_3_ in acetonitrile gave compound **18**. Then click reactions between compound **17** with **18** in the presence of CuSO_4_ and sodium ascorbate in THF/H_2_O (1:1) gave compounds **19a**–**e**. Finally, all the target compounds were fully characterized by NMR and HMRS which was showed in the Appendix A.

### 2.2. Anti-proliferative Activity and Structure Activity Relationships

According to the latest cancer data in China [39] and the actual situation of our laboratory, the in vitro anti-proliferative activity of new target compounds **15a**–**h** and **19a**–**e** were evaluated against MGC-803 cell line (human gastric cancer), HCT-116 cell line (human colon cancer) and PC-3 cell line (human prostate cancer) using MTT assays with the well characterized tubulin inhibitor colchicine was used as a positive control. The following Table 1 and Table 2 depicted the results of in vitro anti-proliferative activity.

To explore the relationships between chemical groups and anti-proliferative activity, compounds **15a**–**h** were designed and synthesized. Representative chemical features of compounds **15a**–**h** are that different moiety such as indoles, indolines and tetrahydroquinoline, which are linked to diarylamide core through an alkyl linker. Most of these compounds exhibited certain anti-proliferative potency against three human cancer cell lines. Particularly, compound **15b** exhibited the most potent inhibitory activity against the tested three human cancer cell lines (MGC-803, PC-3 and EC-109) with IC_50_ values of 1.56 µM, 3.56 μM and 14.5 μM, respectively. The anti-proliferative activity of the compounds varies with its substituent moieties. Compared compounds **15a** with **15g** and **15h**, compounds with an indole group exhibited better anti-proliferative potency than compounds **15g** and **15h** with indoline and tetrahydroquinoline. Similarly, the substituent groups of indole group were also important for anti-proliferative activity. When indole ring is substituted by methoxy group at 5-position, the activity of the compound is the best. However, the anti-proliferative activity of compound was less potent when there was an aldehyde group at the 3-position of indole ring (compounds **15d**, **15e** and **15f**) than that of compounds **15a** and **15b**. In addition, most of compound were more sensitive to MGC-803 cells than to PC-3 and EC-109 cells. These inhibitory results suggested that substituent moieties of compounds exhibited significant effects on anti-proliferative efficacy.

The 1,2,3-Triazole was usually used as a potentially pharmacological linker and fragment to design novel anti-cancer hybrids in medicinal chemistry [40]. Therefore, compounds **19a**–**e** were designed and synthesized by replacing the alkyl linker with a 1,2,3-triazole linker to explore the effects of linkers on activity further. As shown in Table 2, the inhibitory potency of compounds **19a**–**e** was decreased when the alkyl linker was replaced by a 1,2,3-triazole linker (compounds **19a** vs. **15b** and **19b** vs. **15d**), indicating that the 1,2,3-triazole linker could not improve the inhibitory potency for the diarylamide indole derivatives. Besides, compound **19b** with a 5-methoxyindole group also exhibited the strongest inhibitory activity in this series compounds, which was consist with that of compound **15b**. Most of this series compounds were also more sensitive to MGC-803 cells than to PC-3 and EC-109 cells.

Based on the above inhibitory activity results of compounds **15a**–**h** and **19a**–**e**, the structure–activity relationships were summarized. We concluded that nitrogen heterocycle played a significant role in anti-proliferative activity (indole > indoline > tetrahydroquinoline). Proper liner is beneficial to maintain anti-proliferative activity (alkyl linker > 1,2,3-triazole linker).

### 2.3. Compound ***15b*** Inhibited Tubulin Polymerization

As its target protein, inhibitory effects on tubulin polymerization of compound **15b** was first to be detected. Cell free tubulin polymerization assay was performed to evaluate anti-tubulin polymerization activity of compound **15b** at different concentrations with the famous tubulin inhibitors colchicine and paclitaxel. As shown in Figure 4A, when tubulin was incubated with **15b** (10 µM and 20 µM), the increased tendency of the fluorescence intensity was obviously slowed down with a similar action to that of colchicine, which indicated that compound **15b** inhibited tubulin polymerization in a dose-dependent manner. However, the inhibitory activity of compound **15b** on tubulin polymerization is less potent than that of colchicine, which is also consistent with results the anti-proliferative activity. Next, to investigate the effects to microtubules, compound **15b** was selected to do immunofluorescence assay by staining tubulin. As shown in Figure 4B, cells’ morphologies were captured with immunofluorescence (IF) assay. MGC-803 cells treated with **15b** at various concentrations (0.5 µM, 1 µM, and 2 µM) for 24 h resulted in disruption of microtubule networks, while the tubulins were polymerized to micro-tubes in control group. These results indicated that compound **15b** produced the inhibition of tubulin polymerization a dose-dependent manner and caused microtubule network disruption in MGC-803 cells.

### 2.4. Compound ***15b*** Bound to the Colchicine Site of β-tubulin and Molecular Docking Study

The *N*,*N*′-ethylenebis (iodoacetamide) (EBI) assay is usually used to test the binding ability of small molecules to β-tubulin at colchicine binding sites. Therefore, whether compound **15b** acts on the colchicine binding site of tubulin was next to detected. As shown in Figure 5A, the results showed that with the increase of the concentration of compound **15b**, β-tubulin adducts decreased gradually, which indicated that compound **15b** directly bound to the colchicine site of β-tubulin. The Cellular Thermo Shift Assay (CTSA) can detect the direct interaction between compound and proteins. As shown in Figure 5C,D, compound **15b** (100 μM) obviously accelerated the decrease of β-tubulin. These results suggested that compound **15b** directly interacted with β-tubulin and targeted the colchicine binding site.

Compound **15b** displayed inhibitory effects on tubulin polymerization in the screening above, and we then selected it as the optimized compound for the molecular docking studies by Autodock software. To investigate the binding site of compound **15b** with the tubulin-microtubule system, PDB code 1SA0 was selected. The docking results were listed in Figure 5B, trimethoxyphenyl is located in a hydrophobic pocket consist of Val238, Cys241, Leu242, Leu248, Leu259, Ala316, Val319, Ile378 and other residues, and forms a strong hydrophobic interaction with this pocket. 4-methoxyphenyl occupies another hydrophobic pocket and forms extensive hydrophobic interactions with Val191, Met259, Thr314, Ala316, Lys352 and other residues. In addition, the substituted methoxy group on the indole ring faces the vicinity of the electronegative phosphate of the GTP molecule, forming a favorable electrostatic match with this region. At the same time, it also forms a certain hydrophobic effect with residues such as Ala190, Leu248 and Lys254. The above results indicated that compound **15b** bound tightly to the colchicine binding site of β-tubulin and showed polymerization inhibitory activity on β-tubulin.

## 3. Materials and Methods

All the chemical reagents were purchased from commercial suppliers (Energy chemical Company, Shanghai, China and Zhengzhou HeQi Company, Zhengzhou, China). Melting points were determined on an X-5 micromelting apparatus. NMR spectra data was recorded with a Bruker spectrometer. HRMS spectra data was obtained using a Waters Micromass spectrometer. HPLC conditions: injection volume: 10 μL, flow rate, 1 mL/min with a mobile phase of H_2_O/MeOH; H_2_O/MeOH = 55/45 was initially held for 3 min, followed by a linear gradient from 55/45 to 5/95 = H_2_O/MeOH over 15 min, which was then held for 12 min.

### 3.1. Synthesis of Compound ***11***

A solution of commercially available 4-methoxybenzylchloride **9** (1.0 mmol, 1.0 eq), 3,4,5-trimethoxyaniline **10** (1.0 mmol, 1.0 eq), K_2_CO_3_ (2.0 mmol, 2.0 eq) were added into 20 mL DMF, and the reaction was stirred for 6 h at 25 °C. Upon completion, add 15 mL water and extract aqueous layer three times using ethyl acetate (20 mL). The collected organic layer was washed with saturated salt water, dried with magnesium sulfate anhydrous and evaporated to get crude product. The crude product was purified by column chromatography to obtain compound **11**.

### 3.2. Synthesis of Compound ***13***

A solution of compound **11** (1.0 mmol, 1.0 eq) and chloroacetyl chloride **12** (1.5 mmol, 1.5 eq) was added into 20 mL dichloromethane, and the reaction was stirred for 4 h at 25 °C. Upon completion, organic phase was collected to obtain crude products and then were purified with column chromatography to give compound **13**.

### 3.3. Synthesis of Compounds ***15a**–**h***

A solution of compound **13** (1.0 mmol, 1.0 eq), substituted indoles, indolines or tetrahydroquinoline **14** (1.5 mmol, 1.5 eq) and K_2_CO_3_ (2.0 mmol, 2.0 eq) were added were 20 mL acetonitrile, and the reactions were stirred for 8 h at 80 °C. Upon completion, organic phase was collected to obtain crude products and then were purified with column chromatography to give compound **15a**–**h**.

*2-(1H-Indol-1-yl)-N-(4-methoxybenzyl)-N-(3,4,5-trimethoxyphenyl) acetamide* (**15a**). Yield, 47%, m.p. 162–163 °C, White solid. ^1^H NMR (400 MHz, DMSO-*d*_6_) δ 7.52 (d, *J* = 7.6 Hz, 1H), 7.25 (d, *J* = 8.2 Hz, 1H), 7.08 (ddd, *J* = 35.7, 15.9, 8.0 Hz, 5H), 6.86 (d, *J* = 8.1 Hz, 2H), 6.54 (s, 2H), 6.39 (s, 1H), 4.81 (d, *J* = 38.6 Hz, 4H), 3.69 (t, *J* = 12.8 Hz, 12H). ^15^C NMR (101 MHz, DMSO-*d*_6_) δ 167.03, 158.48, 153.04, 157.02, 156.27, 156.08, 129.78, 129.62, 129.25, 127.98, 120.88, 120.16, 118.94, 115.60, 109.70, 105.99, 100.65, 60.00, 55.96, 55.01, 51.86, 47.88, 40.11, 39.90, 39.70, 39.49, 39.28, 39.07, 38.86. HR-MS (ESI): Calcd. C_27_H_28_N2O_5_, [M + H]^+^ *m*/*z*: 461.2071, found: 461.2074. HPLC: *t*_R_ 7.06 min, purity 93.97%.

*2-(5-Methoxy-1H-indol-1-yl)-N-(4-methoxybenzyl)-N-(3,4,5-trimethoxyphenyl) acetamide* (**15b**). Yield, 50%, m.p.: 173–174 °C. ^1^H NMR (400 MHz, DMSO*-d*_6_) δ 7.19–7.09 (m, 4H), 7.03 (d, *J* = 2.3 Hz, 1H), 6.86 (d, *J* = 8.5 Hz, 2H), 6.74 (dt, *J* = 11.9, 6.0 Hz, 1H), 6.52 (s, 2H), 6.30 (d, *J* = 3.0 Hz, 1H), 4.80 (d, *J* = 15.3 Hz, 2H), 4.76 (s, 2H), 3.74 (s, 3H), 3.71 (d, *J* = 8.6 Hz, 9H), 3.65 (s, 3H). ^15^C NMR (100 MHz, DMSO*-d*_6_) δ 198.96, 168.05, 162.86, 156.28, 152.78, 149.30, 159.04, 156.65, 153.86, 129.86, 129.24, 128.61, 125.30, 124.70, 124.15, 120.40, 34.84, 30.79. HR-MS (ESI): Calcd. C_28_H_30_N_2_O_6_, [M + H]^+^ *m*/*z*: 491.2197, found: 491.2180. HPLC: *t_R_* 7.76 min, purity 92.32%.

*2-(6-Methoxy-1H-indol-1-yl)-N-(4-methoxybenzyl)-N-(3,4,5-trimethoxyphenyl) acetamide* (**15c**). Yield, 38%, m.p. 146–147 °C, White solid. ^1^H NMR (400 MHz, DMSO-*d*_6_) δ 7.38 (d, *J* = 8.4 Hz, 1H), 7.14 (d, *J* = 8.4 Hz, 2H), 7.02 (d, *J* = 3.0 Hz, 1H), 6.86 (d, *J* = 8.5 Hz, 2H), 6.71–6.63 (m, 2H), 6.49 (s, 2H), 6.31 (d, *J* = 3.0 Hz, 1H), 4.79 (d, *J* = 18.2 Hz, 4H), 3.75 (s, 3H), 3.72 (s, 3H), 3.68 (s, 6H), 3.64 (s, 3H).^15^C NMR (101 MHz, DMSO-*d*_6_) δ 167.11, 158.49, 155.43, 152.99, 156.97, 156.12, 129.78, 129.33, 128.23, 122.11, 120.68, 115.59, 108.78, 105.82, 100.73, 93.28, 59.94, 55.88, 55.15, 55.02, 51.86, 47.89. HR-MS (ESI): Calcd. C_28_H_30_N_2_O_6_, [M + H]^+^ *m*/*z*: 491.2197, found: 491.2180. HPLC: *t_R_* 6.73 min, purity 93.22%.

*2-(3-Formyl-1H-indol-1-yl)-N-(4-methoxybenzyl)-N-(3,4,5-trimethoxyphenyl) acetamide* (**15d**). Yield, 51%, m.p. 151–152 °C, ^1^H NMR (400 MHz, DMSO-*d*_6_) δ 9.95 (s, 1H), 8.15 (d, *J* = 7.3 Hz, 2H), 7.49 (d, *J* = 7.9 Hz, 1H), 7.38–7.28 (m, 2H), 7.19 (d, *J* = 8.5 Hz, 2H), 6.91 (d, *J* = 8.5 Hz, 2H), 6.63 (s, 2H), 5.07 (s, 2H), 4.83 (s, 2H), 3.77 (s, 9H), 3.72 (s, 3H). White solid. ^15^C NMR (101 MHz, DMSO-*d*_6_) δ 184.63, 166.11, 158.53, 153.10, 142.01, 157.66, 157.12, 155.71, 129.83, 129.06, 124.38, 123.38, 122.35, 120.81, 119.32, 115.61, 111.06, 106.04, 60.00, 55.99, 55.02, 51.99, 48.60. HR-MS (ESI): Calcd. C_28_H_28_N2O6, [M + H]^+^ *m*/*z*: 489.2020, found: 489.2026. HPLC: *t_R_* 5.01 min, purity 98.91%.

*2-(6-Methoxy-1H-indol-1-yl)-N-(4-methoxybenzyl)-N-(3,4,5-trimethoxyphenyl) acetamide* (**15e**). Yield, 38%, m.p. 162–163 °C, White solid. ^1^H NMR (400 MHz, DMSO-*d*_6_) δ 9.86 (s, 1H), 8.03 (s, 1H), 7.59 (d, *J* = 2.3 Hz, 1H), 7.34 (d, *J* = 8.9 Hz, 1H), 7.14 (d, *J* = 8.4 Hz, 2H), 6.96–6.82 (m, 3H), 6.57 (s, 2H), 4.99 (s, 2H), 4.78 (s, 2H), 3.80 (s, 3H), 3.72 (d, *J* = 2.1 Hz, 9H), 3.67 (s, 3H), 3.35 (s, 4H). ^15^C NMR (101 MHz, DMSO-*d*_6_) δ 184.48, 166.12, 158.52, 155.87, 153.08, 141.94, 157.08, 155.70, 152.54, 129.82, 129.06, 125.15, 119.09, 115.60, 115.02, 111.95, 106.00, 102.61, 59.98, 55.97, 55.35, 55.01, 51.99, 48.78. HR-MS (ESI): Calcd. C_29_H_30_N_2_O_7_, [M + H]^+^ *m*/*z*: 519.2126, found: 519.2151. HPLC: *t_R_* 4.90 min, purity 92.22%.

*2-(3-Formyl-5-methyl-1H-indol-1-yl)-N-(4-methoxybenzyl)-N-(3,4,5-trimethoxyphenyl) acetamide* (**15f**). Yield, 33%, m.p. 170–171 °C, White solid. ^1^H NMR (400 MHz, DMSO-*d*_6_) δ 9.86 (s, 1H), 8.03 (s, 1H), 7.90 (s, 1H), 7.31 (d, *J* = 8.3 Hz, 1H), 7.15 (t, *J* = 7.1 Hz, 3H), 6.86 (d, *J* = 8.3 Hz, 2H), 6.57 (s, 2H), 4.98 (s, 2H), 4.77 (s, 2H), 3.69 (d, *J* = 22.0 Hz, 12H), 2.42 (s, 3H).^15^C NMR (101 MHz, DMSO-*d*_6_) δ 184.49, 166.14, 158.52, 153.09, 141.96, 157.10, 156.06, 155.72, 151.39, 129.82, 129.07, 124.77, 124.63, 120.58, 119.01, 115.60, 110.68, 106.01, 59.99, 55.97, 55.01, 51.98, 48.62, 21.07. HR-MS (ESI): Calcd. C_29_H_30_N_2_O_6_, [M + Na]^+^ *m*/*z*: 525.1996, found: 525.2001. HPLC: *t_R_* 5.74 min, purity 90.88%.

*2-(5-Bromoindolin-1-yl)-N-(4-methoxybenzyl)-N-(3,4,5-trimethoxyphenyl) acetamide* (**15g**). Yield, 54%, m.p. 149–150 °C, White solid. ^1^H NMR (400 MHz, DMSO-*d*_6_) δ 7.15–7.01 (m, 4H), 6.86 (d, *J* = 8.5 Hz, 2H), 6.48 (s, 2H), 6.22 (d, *J* = 8.3 Hz, 1H), 4.75 (s, 2H), 3.80 (s, 2H), 3.72 (s, 3H), 3.67 (s, 6H), 3.63 (s, 3H), 3.43 (t, *J* = 8.5 Hz, 2H), 2.88 (t, *J* = 8.4 Hz, 2H).^15^C NMR (101 MHz, DMSO-*d*_6_) δ 167.83, 158.44, 152.97, 151.05, 156.91, 156.58, 151.87, 129.68, 129.49, 129.15, 126.63, 115.61, 107.37, 107.07, 105.84, 60.01, 55.99, 55.02, 52.45, 51.58, 49.22, 27.66. HR-MS (ESI): Calcd. C_27_H_29_BrN_2_O_5_, [M + H]^+^ *m*/*z*: 541.1533, found: 541.1537. HPLC: *t_R_* 10.97 min, purity 95.64%.

*2-(6-Methoxy-3,4-dihydroquinolin-1(2H)-yl)-N-(4-methoxybenzyl)-N-(3,4,5-trimethoxyphenyl) acetamide* (**15h**). Yield, 32%, m.p. 139–140 °C, White solid. ^1^H NMR (400 MHz, DMSO-*d*_6_) δ 7.12 (d, *J* = 8.4 Hz, 2H), 6.85 (d, *J* = 8.5 Hz, 2H), 6.59–6.45 (m, 4H), 6.23 (d, *J* = 8.7 Hz, 1H), 4.74 (s, 2H), 3.83 (s, 2H), 3.70 (d, *J* = 15.4 Hz, 9H), 3.64 (d, *J* = 6.2 Hz, 6H), 3.25–3.18 (m, 2H), 2.64 (t, *J* = 6.1 Hz, 2H), 1.83–1.70 (m, 2H). ^13^C NMR (100 MHz, DMSO-*d*_6_) δ 169.06, 158.40, 153.00, 150.35, 139.44, 136.87, 136.75, 129.65, 129.62, 123.12, 114.73, 113.56, 111.98, 110.86, 105.61, 60.02, 55.99, 55.23, 55.00, 53.19, 51.49, 49.66, 27.58, 21.92. HR-MS (ESI): Calcd. C_29_H_34_N_2_O_6_, [M + H]^+^ *m*/*z*: 507.2490, found: 507.2496. HPLC: *t_R_* 10.97 min, purity 95.64%.

### 3.4. Synthesis of Compounds ***17***

A solution of commercially available 3-bromopropyne **16** (1.5 mmol, 1.0 eq), indolines or tetrahydroquinoline **14** (1.5 mmol, 1.0 eq) and K_2_CO_3_ (2.0 mmol, 1.0 eq) was added into 20 mL acetonitrile, and the reactions were stirred for 6 h at 80 °C. Upon completion, the organic phase was collected to obtain crude products and then were purified with column chromatography to give compounds **17**.

### 3.5. Synthesis of Compounds ***18***

A solution of compound **13** (1.0 mmol, 1.0 eq), sodium azide (2.0 mmol, 2.0 eq) and K_2_CO_3_ (2.0 mmol, 2.0 eq) was added to 20 mL acetonitrile, and the reactions were stirred for 2 h at 80 °C. Upon completion, the organic phase was collected to obtain crude products and then were purified with column chromatography to give compound **18**.

### 3.6. Synthesis of Compounds ***19a**–**e***

A solution of compound **17** (1.0 mmol, 1.0 eq), compounds **18** (1.0 mmol, 1.0 eq), CuSO_4_ (0.1 mmol, 0.1 eq) and sodium ascorbate (0.1 mmol, 0.1 eq) was added to THF/H_2_O (5 mL/5 mL), and the reactions were stirred for 4 h at 25 °C. Upon completion, the organic phase was collected to obtain crude products and then were purified with column chromatography to give compounds **19a**–**e**.

*2-(4-((5-Methoxy-1H-indol-1-yl)methyl)-1H-1,2,3-triazol-1-yl)-N-(4-methoxybenzyl)-N-(3,4,5-trimethoxyphenyl)acetamide* (**19a**). Yield, 47%, m.p. 158–159 °C, White solid. ^1^H NMR (400 MHz, DMSO-*d*_6_) δ 7.85 (s, 1H), 7.46 (d, *J* = 8.8 Hz, 1H), 7.38 (s, 1H), 7.12 (d, *J* = 7.5 Hz, 2H), 7.04 (s, 1H), 6.85 (d, *J* = 7.6 Hz, 2H), 6.77 (d, *J* = 8.9 Hz, 1H), 6.55 (s, 2H), 6.35 (s, 1H), 5.43 (s, 2H), 5.08 (s, 2H), 4.76 (s, 2H), 3.74 (s, 3H), 3.70 (d, *J* = 8.4 Hz, 9H), 3.65 (s, 3H).^15^C NMR (101 MHz, DMSO-*d*_6_) δ 165.65, 159.02, 154.03, 153.55, 143.89, 157.60, 155.94, 151.32, 150.30, 129.52, 129.41, 129.14, 125.15, 114.10, 111.68, 111.24, 106.56, 102.65, 101.09, 60.49, 56.45, 55.79, 55.51, 52.33, 51.50, 41.46. HR-MS (ESI): Calcd. C_31_H_33_N_5_O_6_, [M + H]^+^ *m*/*z*: 572.2504, found: 572.2509. HPLC: *t_R_* 5.29 min, purity 92.17%.

*2-(4-((2-Formyl-1H-indol-1-yl)methyl)-1H-1,2,3-triazol-1-yl)-N-(4-methoxybenzyl)-N-(3,4,5-trimethoxyphenyl)acetamide* (**19b**). Yield, 47%, m.p. 168–169 °C, White solid. ^1^H NMR (400 MHz, DMSO-*d*_6_) δ 9.94 (s, 1H), 8.41 (s, 1H), 8.15–8.02 (m, 2H), 7.73 (d, *J* = 8.0 Hz, 1H), 7.29 (dd, *J* = 15.6, 7.5 Hz, 2H), 7.12 (d, *J* = 8.1 Hz, 2H), 6.85 (d, *J* = 8.2 Hz, 2H), 6.56 (s, 2H), 5.64 (s, 2H), 5.11 (s, 2H), 4.77 (s, 2H), 3.69 (t, *J* = 12.0 Hz, 12H).^15^C NMR (101 MHz, DMSO-*d*_6_) δ 185.20, 165.63, 159.03, 153.58, 142.49, 141.11, 157.64, 157.30, 155.93, 150.32, 129.40, 125.78, 125.21, 124.08, 123.07, 121.52, 119.90, 114.10, 111.84, 106.57, 60.49, 56.45, 55.50, 52.37, 51.62, 42.04. HR-MS (ESI): Calcd. C_31_H_31_N_5_O_6_, [M + H]^+^ *m*/*z*: 592.2167, found: 592.2192. HPLC: *t_R_* 4.41 min, purity 95.57%.

*N-(4-Methoxybenzyl)-2-(4-((2-methylindolin-1-yl)methyl)-1H-1,2,3-triazol-1-yl)-N-(3,4,5-trimethoxyphenyl)acetamide* (**19c**). Yield, 39%, m.p. 171–172 °C, ^1^H NMR (400 MHz, DMSO-*d*_6_) δ 7.78 (s, 1H), 7.12 (d, *J* = 8.5 Hz, 2H), 6.95 (dt, *J* = 7.2, 3.7 Hz, 2H), 6.85 (d, *J* = 8.6 Hz, 2H), 6.56 (dd, *J* = 15.9, 7.6 Hz, 4H), 5.07 (s, 2H), 4.76 (d, *J* = 2.5 Hz, 2H), 4.52 (d, *J* = 16.0 Hz, 1H), 4.25 (d, *J* = 15.9 Hz, 1H), 3.70 (d, *J* = 7.1 Hz, 9H), 3.65 (s, 3H), 3.02 (dd, *J* = 15.5, 8.4 Hz, 1H), 1.31 (d, *J* = 6.1 Hz, 3H). ^15^C NMR (101 MHz, DMSO-*d*_6_) δ 165.73, 159.02, 153.57, 152.02, 143.18, 157.61, 156.00, 150.29, 129.44, 129.02, 127.53, 125.22, 124.36, 119.69, 114.10, 107.60, 106.56, 60.49, 59.08, 56.45, 55.50, 52.32, 51.45, 37.08, 19.26. HR-MS (ESI): Calcd. C_31_H_35_N_5_O_5_, [M + H]^+^ *m*/*z*: 558.2711, found: 558.2715. HPLC: *t_R_* 5.19 min, purity 96.55%.

*2-(4-(Indolin-1-ylmethyl)-1H-1,2,3-triazol-1-yl)-N-(4-methoxybenzyl)-N-(3,4,5-trimethoxyphenyl) acetamide* (**19d**). Yield, 51%, m.p. 165–166 °C, White solid. ^1^H NMR (400 MHz, DMSO-*d*_6_) δ 7.78 (s, 1H), 7.12 (d, *J* = 8.5 Hz, 2H), 6.95 (dt, *J* = 7.2, 3.7 Hz, 2H), 6.85 (d, *J* = 8.6 Hz, 2H), 6.56 (dd, *J* = 13.9, 7.6 Hz, 4H), 5.07 (s, 2H), 4.76 (d, *J* = 2.5 Hz, 2H), 4.52 (d, *J* = 16.0 Hz, 1H), 4.25 (d, *J* = 15.9 Hz, 1H), 3.70 (d, *J* = 7.1 Hz, 9H), 3.65 (s, 3H), 3.02 (dd, *J* = 15.5, 8.4 Hz, 1H), 1.31 (d, *J* = 6.1 Hz, 3H). ^13^C NMR (101 MHz, DMSO-*d*_6_) δ 165.23, 158.54, 153.08, 151.49, 143.29, 142.62, 157.18, 155.51, 129.80, 129.75, 128.94, 127.00, 124.88, 124.20, 119.38, 115.62, 107.39, 106.10, 60.01, 55.99, 55.02, 52.34, 51.85, 50.97, 43.19, 27.87. HR-MS (ESI): Calcd. C_30_H_33_N_5_O_5_, [M + H]^+^ *m*/*z*: 544.2554, found: 544.2559. HPLC: *t_R_* 7.1 min, purity 88.75%.

N-(4-Methoxybenzyl)-2-(4-((6-methyl-3,4-dihydroquinolin-1(2H)-yl)methyl)-1H-1,2,3-triazol-1-yl)-N-(3,4,5-trimethoxyphenyl)acetamide (**19e**). Yield, 44%, m.p. 162–163 °C, 1H NMR (400 MHz, DMSO-d_6_) δ 7.76 (s, 1H), 7.15 (d, J = 8.2 Hz, 2H), 6.85 (d, J = 8.2 Hz, 2H), 6.75–6.61 (m, 3H), 6.55 (s, 2H), 5.07 (s, 2H), 4.77 (s, 2H), 4.47 (s, 2H), 3.70 (d, J = 7.0 Hz, 9H), 3.65 (s, 3H), 3.31–3.26 (m, 2H), 2.62 (t, J = 6.0 Hz, 2H), 2.10 (s, 3H), 1.90–1.82 (m, 2H). ^13^C NMR (101 MHz, DMSO-d_6_) δ 165.75, 159.02, 153.56, 144.07, 142.98, 157.60, 156.00, 150.29, 150.00, 129.44, 127.64, 124.74, 124.57, 122.68, 114.10, 111.87, 106.55, 60.49, 56.45, 55.50, 52.32, 51.44, 49.37, 46.44, 27.89, 22.42, 20.37. HR-MS (ESI): Calcd. C_32_H_35_N_5_O_5_, [M + Na]^+^ m/z: 594.2687, found: 594.2691. HPLC: t_R_ 9.50 min, purity 86.08%.

### 3.7. Cell Culture

Cells were cultured in RPMI-1640 medium supplemented with 10% fetal bovine serum (FBS), 100 U/mL penicillin and 0.1 mg/mL streptomycin. All the cells were incubated at 37 °C and 5% CO_2_.

### 3.8. MTT Assay

A total of 5000 cells were seeded into 96-well cell culture plates. After 24 h, cells were treated with synthesized compounds. Then, MTT reagent was added 20 μL per well after 48 h treatment with synthesized compounds. Cells were then incubated for 4 h at 37 °C. Formazan was then dissolved with DMSO. Absorbencies of formazan solution at 490 nm were determined. SPSS version 10.0 was used for 50% inhibitory concentration (IC_50_) calculation [41,42].

### 3.9. Tubulin Polymerization Detection

Pig brain microtubule protein was isolated by three cycles of temperature-dependent assembly/disassembly in PIPES (pH 6.5, 100 mM), MgSO_4_ (1.0 mM), EGTA (2.0 mM), GTP (1.0 mM) and 2-mercaptoethanol (1.0 mM). In the first cycle of polymerization, glycerol and phenylmethylsulfonyl fluoride were added to 4 M and 0.2 mM, respectively. Homogeneous tubulin was prepared from microtubule protein by phosphocellulose (P11) chromatography. The purified proteins were stored in aliquots at −70 °C.

Re-suspend tubulin in proton exchange membrane buffer (containing 100 mmol/L PIPES, 1 mmol/L EGTA, 0.5 mmol/L Mgcl_2_, 1 mmol/L GTP, 10.2% glycerol), and the solution was incubated with different concentrations of compound **15b** (10, 20 μmol/L), colchicine (3 μmol/L), paclitaxel (3 μmol/L) and the carrier DMSO on ice. Using a spectrophotometer to monitor the absorbance of the reaction at 420 nm (excitation wavelength is 340 nm) [43].

### 3.10. Cellular Thermal Shift Assay

Inoculate the cells in a petri dish, and collect the cells when the cells grow to 90%. The cells were re-suspended in PBS containing phosphatase inhibitor and protease inhibitor, and the cells were repeatedly frozen and thawed in liquid nitrogen. Compound **15b** (100 μmol/L) and the same amount of DMSO were added to the protein respectively, and the mixture was heated at 50 °C, 55 °C, 60 °C, 65 °C and 70 °C for three times. The supernatant was centrifuged at 15,000× *g* rpm, and the sample was used for western blot analysis.

### 3.11. EBI Competition Experiment

In total, 3 × 10^5^ MGC-903 cells ells were seeded into 6-well cell culture plates and cultured for 24 h, incubated with different concentrations of compound **15b**, colchicine, paclitaxel, and DMSO for 2 h, and then treated with 100 μmol/L EBI for 1.5 h. The cells were collected, and the β-tubulin and β-tubulin adducts were determined with anti-β-tubulin antibody.

### 3.12. Immunofluorescence Experiment

MGC-803 cells were seeded in 96-well plates and treated with different concentrations of compound **15b** for 24 h. Fix the cells with 4% paraformaldehyde for 10 min, and then infiltrate the cells with PBS containing 0.1% Triton X-100. After blocking with 5% BSA at room temperature for 1 h, incubate overnight with anti-β-tubulin antibody at 4 °C, stain with fluorescent antibody, and label cell nuclei with DAPI. Observe the cells using a fluorescence microscope.

### 3.13. Molecular Docking

The molecular docking study was performed using MOE 2015.10. The crystal structure of tubulin (PDB ID: 1SA0) was retrieved from RCSB Protein Data Bank (https://www.rcsb.org/structure/1SA0, accessed on 1 July 2021), and then was prepared by adding hydrogen atoms, removing water molecules and repairing the missing side chains. The protonation states of protein residues were calculated in the pKa at 7. The ligand compound **15b** was built in Autodock software (Scripps Research Institute, La Jolla, CA, USA) and was prepared by energy minimization and conformational search. The ligand was docked into the colchicine binding site of tubulin and 20 poses were exported for the next analysis.

## 4. Conclusions

Tubulin has been regarded as an attractive and successful molecular target in cancer therapy and drug discovery. However, owing to the poor water solubility, drug resistance and side effects of clinical tubulin inhibitors, it is necessary to develop novel tubulin inhibitors. As the continuation of our group work on colchicine binding-site tubulin inhibitors, we designed and synthesized a series of diarylamide indole derivatives by the combination of vicinal diaryl core and indole skeleton into one hybrid though proper linkers. Among of these compounds, compound **15b** exhibited the most potent inhibitory activity against the tested three human cancer cell lines (MGC-803, PC-3 and EC-109) with IC_50_ values of 1.56 µM, 3.56 μM and 14.5 μM, respectively. Besides, the SARs of these compounds were preliminarily studied and summarized. The most active compound **15b** produced the inhibition of tubulin polymerization in a dose-dependent manner and caused microtubule network disruption in MGC-803 cells. Therefore, compound **15b** was identified as a novel tubulin polymerization inhibitor targeting the colchicine binding site. In addition, the results of molecular docking also suggested compound **15b** tightly bind into the colchicine binding site of β-tubulin.

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
