# Peer review of "Discovery of Novel Diarylamide N-Containing Heterocyclic Derivatives as New Tubulin Polymerization Inhibitors with Anti-Cancer Activity"

_molecules, 2021, doi:10.3390/molecules26134047_

Round 1
Reviewer 1 Report
The article of Liu et al. deals with novel inhibitors of tubulin polymerization with potential anticancer activity. Authors synthetized several diacrylamine N-containing heterocyclic derivatives and tested their effect to tubulin polymerization in in vitro assays as well as the effect of microtubules on cellular level. Binding to tubulin as well as antiproliferation aktivity were tested.
Part of manuscript dealing with synthesis of novel compounds is described in sufficient details and I do not have critical comments. On the other hand, part of the manuscript focused to biological effects of the drugs requires substantial changes and improvements.
Testing an effect of the drugs to tubulin polymerization should be described in more details in sections of Material and Methods, Results as well as in Discussion. For example data on tubulin used in the assay are missing, was it purified bovine or porcine tubulin, or other tubulin?. As a quality of tubulin is prerequisite of the assay, supplier and cat no. should be provided.
Sufficient information on testing the drugs for anti-proliferating activity is not given either in MM, or in the text of Results and text to Table 1 or 2 .
More information should be provided in addition to the sentence:
3.12. General Methods
In this work, some other assays including cell culture assay and MTT assay were referred to our previous work [33, 34].
Link to database and software used for molecular docking study is missing.
Size of Fig. 5 B should be enlarged, description of AA residues is not readable due to a small letter size. Description of the Figure 5B and in general Text to all Figures should provide readers with more information on presented data.
Fig. 5C, D instead compd., the specification of the compound 15d should be provided in the both figures.
English has to be corrected to make the submitted manuscript publishable. Just one example of an incorrect scientific English which is often used in the paper. The sentence using MTT assays with the famous tubulin inhibitor Colchicine as a positive drug. should be corrected Well characterized (established) tubulin inhibitor Colchicine was used as a positive control.
Author Response
1.Testing an effect of the drugs to tubulin polymerization should be described in more details in sections of Material and Methods, Results as well as in Discussion. For example, data on tubulin used in the assay are missing, was it purified bovine or porcine tubulin, or other tubulin? As a quality of tubulin is prerequisite of the assay, supplier and cat no. should be provided.
Response: Thanks for your valuable comments. We feel sorry for our carelessness and have added descriptions of this part of the work in the revised manuscript.
- In regards to Sufficient information on testing the drugs for anti-proliferating activity is not given either in MM, or in the text of Results and text to Table 1 or 2.
Response: Thanks for your valuable comments. We feel sorry for our carelessness and have corrected it according to your valuable comments in the revised manuscript.
- In regards to General Methods
Response: Thanks for your valuable comments. We feel sorry for our carelessness and have corrected it according to your valuable comments in the revised manuscript.
- Link to database and software used for molecular docking study is missing.
Response: Thanks for your valuable comments. We feel sorry for our carelessness and have corrected it according to your valuable comments in the revised manuscript.
- Size of Fig. 5 B should be enlarged, description of AA residues is not readable due to a small letter size. Description of the Figure 5B and in general Text to all Figures should provide readers with more information on presented data.
Response: Thanks for your valuable comments. We feel sorry for our carelessness and have corrected it according to your valuable comments in the revised manuscript.
- Fig. 5C, D instead compd., the specification of the compound 15d should be provided in the both figures.
Response: Thanks for your valuable comments. We feel sorry for our carelessness and have corrected it according to your valuable comments in the revised manuscript.
- English has to be corrected to make the submitted manuscript publishable. Just one example of an incorrect scientific English which is often used in the paper.
Response: Thanks for your valuable comments. We feel sorry for our carelessness and have corrected the grammar mistakes in the revised manuscript.
Reviewer 2 Report
Please see attached file

Author Response
- It would be helpful if the abstract somehow summarized more thoroughly the series of molecules and provided the tubulin data for compound 15b, along with the provided cell.
Response: Thanks for your valuable comments. We feel sorry for our carelessness and have corrected it according to your valuable comments in the revised manuscript.
- Introduction presumably the authors intended to state " plays a vital role rather than vira role.
Response: Thanks for your valuable comments. We feel sorry for our carelessness and have corrected the mistake in the revised manuscript.
- Introduction: Tubulin has been. rather than " Tubulin have been
Response: Thanks for your valuable comments. We feel sorry for our carelessness and have corrected the mistake in the revised manuscript.
- The introduction is fairly sparse and could certainly be further developed
Response: Thanks for your valuable comments. We feel sorry for our carelessness and have corrected it according to your valuable comments in the revised manuscript.
- There are a variety of sentence structure, grammar, and related issues throughout the manuscript that should be addressed. This reviewer will not attempt to address them all but as one further example in the introduction (highly inhibitory potency " and " could potently inhibits tubulin should be revised.
Response: Thanks for your valuable comments. We feel sorry for our carelessness and have corrected the mistake in the revised manuscript.
- The authors refer to instability of the Z double bond inherent to CA4 as a factor that limited clinical development, however this reviewer is not aware of any clear data that indicates that double-bond isomerization actually limited the clinical development ofCA4. This should be revised, or clearly pertinent literature citations should be provided.
Response: Thanks for your valuable comments. We feel sorry for our carelessness and have provided pertinent literature citations according to your valuable comments in the revised manuscript.
- Similar issue (as mentioned in # 6 above) in regard to mention in poor water-solubility and low bioavailability as clinically limiting factors associated with CA4. The authors should be aware (and should point out) that the clinical work with CA4 was carried out with the corresponding water-soluble phosphate prodrug salt (CA4P). This information should be incorporated within the manuscript.
Response: Thanks for your valuable comments. We feel sorry for our carelessness and have added related information according to your valuable comments in the revised manuscript.
8.This sentence is unclear " " Cushman. modification ethylene bridge of the led CA-4
Response: Thanks for your valuable comments. We feel sorry for our carelessness and have corrected it according to your valuable comments in the revised manuscript.
- The authors have made an attempt to capture the vast repertoire of CA4 modifications known around the ethylene bridge (and provided Fig. 1) - however this area is so vast that additional citations-perhaps citations to pertinent review articles on these bridge (and related) modifications to CA4 is warranted
Response: Thanks for your valuable comments. We feel sorry for our carelessness and have provided additional citations according to your valuable comments in the revised manuscript.
- The authors cite indole-based molecules, but have inadvertently omitted citations to some key contributions in this area of indole-based inhibitors of tubulin polymerization. A literature search should be conducted to identify and cite other important contributions in regard to indole-based inhibitors of tubulin polymerization. Some further coverage of pertinent literature related to indole-based inhibitors of tubulin polymerization seems warranted.
Response: Thanks for your valuable comments. We feel sorry for our carelessness and have provided additional citations according to your valuable comments in the revised manuscript.
11.While compounds 15a-h and 19a-g incorporate functional group diversity and thus add to knowledge gained from overall structure-activity relationship (SAR) studies, all of these compounds lack the phenolic moiety on the CA4 ring that bears the para-methoxy group This seems like either an unfortunate oversight as having molecules in these new series that exactly mimic the two aryl rings of CA4 would add foundational strength to the SAR associated with the ethylene bridge modification. or perhaps the authors considered this but had some synthetic challenges accessing these molecules. It would seem beneficial to include this key phenolic moiety in at least a subset of the new molecules.
Response: Thanks for your valuable comments. In fact, we are working on the synthesis of these molecules including this key phenolic moiety.
- Scheme 1: conditions " d lack a source of azide (such as NAN3) , although it is mentioned in the text . In addition. the reaction between compound 14 and 16 to prepare 17 seems to involve the RI and R2 groups on compound 14, switching to R3 and R4 groups on compound 17 (yet no chemistry seems to involve these groups) , which then switch back to RI and R2 in the compound of the 19-series . This is confusing and likely involves some error in assigning R groups properly.
Response: Thanks for your valuable comments. We feel sorry for our carelessness and have corrected the mistakes in the revised manuscript.
- The boxes at both the top and bottom of Scheme I list R2 and R3 but no molecules displayed in Scheme I contain R2 and R3, rather the molecules contain R and R2. This adds confusion.
Response: Thanks for your valuable comments. We feel sorry for our carelessness and have corrected the mistakes in the revised manuscript.
- Some further description of the three human cancer cell lines used should be provided at least mentioning the type of cancer represented by each cell line and why this subset of cell lines was selected.
Response: Thanks for your valuable comments. We feel sorry for our carelessness and have added related description according to your valuable comments in the revised manuscript.
15.The premise for using the 1, 2, 3-triazole spacer is not well described
Response: Thanks for your valuable comments. We feel sorry for our carelessness and have added related description according to your valuable comments in the revised manuscript.
- It seems pertinent to have IC50 values for inhibition of tubulin polymerization for all of the key compounds. At this point, the authors provide Figure 4 data for only 1 compound15b).
Response: Thanks for your valuable comments. Usually, in many literatures, only the most active compound was tested the inhibitory effects on tubulin polymerization, therefore, we provide Figure 4 data for only compound 15b.
- While the authors indicate that an EBI assay is " usually used " to evaluate colchicine site binding, it should be kept in mind that certain other laboratories use a direct competition assay with radiolabeled colchicine. It is not clear if the EB assay confirms colchicine site binding or just binding to beta-tubulin. This should be clarified.
Response: Thanks for your valuable comments. Usually, in many literatures, The EBI assay is usually used to test the binding ability of small molecules to β - tubulin at colchicine binding sites. In addition, we have added related citations in the revised manuscript.
- The synthetic chemistry details are somewhat sparse. It would be helpful to include g (or mg) amounts used in addition to mmols. Also, it would seem that the measured value ingrams (or mgs) would have more significant figures then presented for the mmol numbers. Presumably " EA ' is an abbreviation for ethyl acetate but this should be clarified. The amount of each key product or intermediate should be reported (g, mmol)
Response: Thanks for your valuable comments. We feel sorry for our carelessness and have added related description according to your valuable comments in the revised manuscript.
- NMR and HRMS data are presented as evidence of structure, however no data (such as HPLC or combustion) is presented as evidence of compound purity. This is especially important for all compounds for which biological data are reported.
Response: Thanks for your valuable comments. We feel sorry for our carelessness and have added HPLC date according to your valuable comments in the revised manuscript.
- Was the tubulin assay done in duplicate or other?
Response: Thanks for your valuable comments. the tubulin assay done in duplicate for three times.
- Details regarding cell culture and the MTT assay should be presented rather than just referencing previous work. Was the cell line date obtained in triplicate?
Response: Thanks for your valuable comments. We feel sorry for our carelessness and have added related description according to your valuable comments in the revised manuscript. In addition, the cell line date obtained in triplicate.
- The authors should review for verification and then confirm that all molecules reported are new molecules, not previously reported. Likely this is the case, but some clear statement would be helpful.
Response: Thanks for your valuable comments. We feel sorry for our carelessness and have added it according to your valuable comments in the revised manuscript.
- The conclusion should be revised to address some of the issues mentioned at various points in this review.
Response: Thanks for your valuable comments. We feel sorry for our carelessness and have corrected it according to your valuable comments in the revised manuscript.
26.The addition of a Table of Contents to the Supplementary Data file would be helpful
Response: Thanks for your valuable comments. We feel sorry for our carelessness and have added a Table of Contents to the Supplementary Data file according to your valuable comments in the revised manuscript.